# Determinants of blood pressure and blood glucose control in patients with co-morbid hypertension and type 2 diabetes mellitus in Ghana: A hospital-based cross-sectional study

**Yakubu Alhassan**[1], **Adwoa Oforiwaa Kwakye**[2]\*, **Andrews K. Dwomoh**[2], **Emmanuella Baah-Nyarkoh**[2], **Vincent Jessey Ganu**[3], **Bernard Appiah**[4], **Irene A. Kretchy**[2]

1 Department of Biostatistics, School of Public Health, University of Ghana, Accra, Ghana, 2 Department of Pharmacy Practice and Clinical Pharmacy, School of Pharmacy, College of Health Sciences, University of Ghana, Legon, Ghana, 3 Department of Internal Medicine, Korle Bu Teaching Hospital, Accra, Ghana, 4 Department of Public Health, Falk College, Syracuse University, Syracuse, NY, United States of America

\* aokwakye@ug.edu.gh

## Abstract

Hypertension and diabetes are major risk factors for cardiovascular diseases and optimal control of blood pressure (BP) and blood glucose are associated with reduced cardiovascular disease events. This study, therefore, sought to estimate the prevalence and associated factors of controlled BP and blood glucose levels among patients diagnosed with both hypertension and Type 2- diabetes mellitus (T2DM). A quantitative cross-sectional study was conducted in a primary health setting in Ghana among patients 18 years and older diagnosed with both hypertension and T2DM. Pearson's chi-square was used to assess the association between BP and blood glucose levels and the independent variables. The multivariable binary logistic regression model was used to assess the adjusted odds of controlled BP and blood glucose levels. Among the 329 participants diagnosed with both hypertension and T2DM, 41.3% (95% CI: 36.1–46.8%) had controlled BP, 57.1% (95% CI: 51.7–62.4%) had controlled blood glucose whilst 21.8% (95% CI: 17.7–26.7%) had both controlled BP and blood glucose levels. Increased age, non-formal education, non-married, employed, single-dose anti-hypertensives or anti-diabetic medications, and hyperlipidaemia or stroke co-morbidities were positively associated with controlled BP levels. Being female, married, taking 2 or more anti-hypertensive medications, and moderate to high medication-related burden were positively associated with controlled blood glucose levels. In terms of both controlled BP and blood glucose levels, being employed, reduced income level, being registered with national health insurance, single anti-diabetes or anti-hypertensive medications, hyperlipidaemia or stroke co-morbidities, and moderate to high medication-related burden were positively associated with having both controlled BP and blood glucose levels. One in five patients with hypertension and T2DM had both BP and blood glucose levels under control. The benefits and risks of blood pressure and blood glucose targets should thus be factored into the management of patients with hypertension and T2DM.

**Data Availability Statement:** The data that support the findings of this study is available in Mendeley data at https://dx.doi.org/10.17632/stb99vj4fb.1.

**Funding:** The author(s) received no specific funding for this work.

**Competing interests:** The authors have declared that no competing interests exist.

**Abbreviations:** BP, Blood pressure; T2DM, Type-2 diabetes mellitus; FBG, Fasting blood glucose; JNC; AOR, adjusted odds ratio; CI, confidence interval; IQR, interquartile range; SBP, systolic blood pressure; DBP, diastolic blood pressure; LMQ, Living with medicines questionnaire.

## Introduction

Hypertension and diabetes remain major public health threats worldwide [1,2]. When these cardio-metabolic conditions co-exist in an individual, there is a worsening of both glycaemic and cardiovascular endpoints [3,4]. Both diseases have similarities in risk factors including life-style, dyslipidaemia, familial, and racial as well as complications [5,6]. These complications can be categorized into micro and macrovascular complications. Myocardial infarction, stroke, peripheral vascular disease, coronary artery disease, and congestive heart failure are examples of macrovascular problems, while retinopathy, nephropathy, and neuropathy are examples of microvascular complications [7,8].

A significant contributor to the onset and development of diabetes-related illnesses and complications is inadequate and poor glycaemic control. This can significantly increase medical costs, diminish the quality of life, and reduce life expectancy [4]. Research has shown that improving glycaemic control can help patients live longer, have an improved quality of life, and delay the development and progression of diabetic complications [9]. Also, improving glycaemic control significantly lowers expenditures associated with the management of diabetes [10]. Blood glucose levels must be tightly controlled to delay the onset of diabetes and its related complications.

Clinical recommendations in diabetes care support have varied blood pressure (BP) values; nonetheless, most advocate lower levels compared to people without diabetes [11,12]. The lowering of BP in patients with diabetes results in a significant reduction in cardiovascular problems [5]. The target BP for diabetic patients should be less than 140/90 mmHg, according to the Joint National Committee's eighth report (JNC 8), and most patients will need to take two or more antihypertensive drugs to do so [12]. Increased patient and healthcare provider understanding about the illness, access to care, suitable lifestyle changes, evidence-based treatment, high levels of medication adherence, and thorough follow-up are all components of high-quality BP control [13–15]. In addition, studies have shown that old age, chronic renal diseases, longer duration of hypertension, and uncontrolled diabetes mellitus are significant risk factors for poor BP control [16,17].

According to a Ghana Ministry of Health report, diabetes and hypertension are among the top fifteen causes of outpatient visits [18,19]. Effective management of these patient groups requires substantive knowledge of the patient characteristics and other factors affecting their BP control. Even though there have been studies on factors associated with blood pressure in diabetes [20,21]. Literature on factors associated with regulated BP as well as controlled blood glucose levels among persons with co-morbid hypertension and diabetes is limited. This study, therefore, assessed the factors associated with BP and blood glucose control among patients with co-morbid hypertension and type 2 diabetes mellitus (T2DM) at an outpatient department of a lower-middle-income country hospital.

## Methods

### Study design and context

A hospital-based cross-sectional study was conducted at the Adabraka Polyclinic which is a public primary health facility in Ghana. The clinic provides all essential general healthcare services including assessment and management of chronic conditions such as hypertension and diabetes mellitus. All the commonly prescribed antihypertensive and oral hypoglycaemic agents are included in the national essential drugs list and covered by the National Health Insurance Scheme (NHIS) [22]. This is in line with the national policy on non-communicable diseases (NCDs) which provides guidelines on the primary prevention (e.g., health

promotion), secondary and tertiary prevention (e.g., screening, and early detection, clinical care/case management) of NCDs [19]. The Outpatient Department (OPD) holds twice-weekly clinics for patients with hypertension and T2DM and provides services for an average of eighty patients daily.

## Study participants

The study engaged and recruited adults aged at least 18 years known to have both hypertension and T2DM with documented evidence of these clinical diagnoses in their clinical files. Patients who were on anti-hypertension and anti-diabetes medications for at least six months prior to the study were included in the study. Participants excluded from the study include those with Type 1 diabetes mellitus, gestational diabetes, and evidence of impaired cognitive function. Participants provided informed consent prior to their inclusion in the study.

This study is part of a previous study among people with co-morbid hypertension and T2DM, with a sample size of 326 estimated, which was based on a 74.05% prevalence rate of medication adherence [4], 95% confidence interval, 5% error margin and a 10% non-response rate. The study participants were recruited systematically between October 2021 and November 2021.

Patients' medical records were reviewed to validate the diagnosis of hypertension and T2DM diagnosis.

## Measures

Data were collected using a validated questionnaire that included general information on socio-demographic characteristics (e.g., age, sex, marital status, religion, education level, occupation, monthly income, payment method for drugs, monthly expenditure), clinical characteristics (number of medications, frequency of daily dose of medication, presence of co-morbidities, blood glucose level, systolic blood pressure (SBP), diastolic blood pressure (DBP), duration since diagnosis and frequency of follow-up visits, and family history of hypertension and T2DM and medication burden.

The patient's BP was measured using a manual mercury sphygmomanometer. Patients were instructed to rest for 3 to 5 minutes before having their BP measured. They sat back in a chair, their back supported. The nurses ensured that the appropriate cuff size was used for BP measurement based on the patient's arm diameter. The readings were taken three times and the average was taken. In this study, participants with SBP below 140mmHg and DBP below 90mmHg were considered to have controlled BP level.

The fasting blood sugar was measured using a certified automated glucometer (GOLD--ACCU). The finger surface was first cleaned with 75% alcohol, and then the sterilised needle was used to prick it. The very first drop of blood was discarded. After 5 seconds, the reading appeared after the blood sample was placed on the strip. These tasks were carried out by well-trained nurses. In this study participants with blood glucose level below 7.0mmol/L were considered to have controlled blood glucose level.

Pill burden was assessed with the 41-item Living with Medicines Questionnaire (LMQ-3) with the overall score ranging from 41 to 205 [23,24]. Pill burden was categorized according to score range as no burden at all (scores in range 41–73); minimum burden (scores in range 74–106); moderate burden (scores in range 107–139); high burden (scores in range 140–172); and Extremely high burden (scores in range 173–205). The LMQ-3 was dichotomized for analysis in this study as moderate/high burden against minimum burden. The scale is reliable with a Cronbach's alpha score 0.9208 computed in this study.

The 5-item medication adherence report scale (MARS-5) measured adherence behaviour [25]. Each item is rated from 1 = always to 5 = never with the composite score ranging from 5 to 25. This study reports a Cronbach's alpha score of 0.8568 for the MARS-5.

## Statistical analysis

Stata IC version 16 (StataCorp, College Station, TX, US) was used for analysis. Descriptive statistics were presented using frequency and percentages for categorical variables, the mean and standard deviation for normally distributed continuous variables, and median and interquartile range (IQR) for non-normally distributed continuous variables. The distribution of SBP, DBP, and blood glucose concentration was described using the box and whiskers plots. The percentage of participants with controlled and uncontrolled levels of BP, blood glucose level and both were presented and corresponding 95% confidence interval were estimated using the binomial exact estimation approach.

The Pearson's chi-square test was used to assess the bivariate association between the socio-demographic, clinical and medication-related characteristics observed in the study and the BP and blood glucose levels among the participants. The multiple binary logistic regression model with robust standard errors was used to assess the adjusted odds of controlled BP, controlled blood glucose and controlled levels of both BP and blood glucose among participants across the various characteristics.

Multicollinearity between the variables was assessed using the variance inflation factor (VIF) which recorded a mean VIF of 3.01 (range: 1.72 to 6.98) which is within the acceptable range of less than 10 all three models. The area under the receiver operating characteristics curve was 0.8552 (95% CI: 0.8130 to 0.8975) for the BP level model, 0.7884 (95% CI: 0.7390 to 0.8376) for the blood glucose model and 0.8537 (95% CI: 0.8041 to 0.9033) for both controlled BP and blood glucose model which were all above 70%. The Hosmer-Lemeshow goodness of fit test was insignificant for both the BP level (p-value = 0.073), blood glucose level (p-value = 0.268) and both BP and blood glucose (p-value = 0.636) model indicating models were appropriately fitted. All statistical analysis in this study were considered significant with p-values less than 0.05.

## Ethics statement

The study was approved by Ghana Health Service Ethical Review Committee (GHS-ERC 043/09/21). Participants were chosen for the study depending on their willingness to participate. Informed consent for participation was also obtained from each patient.

## Results

### Background characteristics of study participants

The mean age of the 329 study participants was 57.5 ± 13.2 years. More than half (56.2%) of them were female and 58.4% were married. Less than a fifth (17.9%) had no formal education whilst 21.3% had a tertiary level of education. The median monthly expenditure on medication was 50.0 cedis (thus, 8.00 USD) with a fifth (20.7%) paying for all their medication using the health insurance. Family history of hypertension and diabetes were 68.1% and 49.8% respectively. Most participants (94.2%) of the participants were on amlodipine as an anti-hypertensive medication. The median number of antihypertensive medications taken was 2 (IQR: 1 to 3) with 70.2% of the participants taking at least 2 different medications. The median number of anti-diabetic medications taken was 2 (IQR: 1 to 2) with 51.1% on at least 2 medications (Table 1).

**Table 1. Background characteristics of study participants (N = 329).**

| Characteristics | Frequency (%) |
| --- | --- |
| **Overall** | 329 (100.0) |
| **SOCIO-DEMOGRAPHICS** | |
| **Sex** | |
| Male | 144 (43.8) |
| Female | 185 (56.2) |
| **Age, Mean [±SD]** | 57.5 [±13.2] |
| **Age group** | |
| <50 | 80 (24.3) |
| 50–59 years | 108 (32.8) |
| 60–69 | 78 (23.7) |
| 70+ | 63 (19.1) |
| **Marital status** | |
| Single | 84 (25.5) |
| Married | 192 (58.4) |
| Divorced | 25 (7.6) |
| Others | 28 (8.5) |
| **Highest education** | |
| No formal education | 59 (17.9) |
| Primary | 72 (21.9) |
| Secondary | 128 (38.9) |
| Tertiary | 70 (21.3) |
| **Occupation** | |
| Unemployed | 41 (12.5) |
| Trader/artisan | 176 (53.5) |
| Professional | 53 (16.1) |
| Retired | 48 (14.6) |
| Others | 11 (3.3) |
| **Monthly income** | |
| 0–500 cedis (0–80 USD) | 150 (45.6) |
| 501–1000 cedis (81.0–160 USD) | 121 (36.8) |
| >1000 (>160 USD) | 58 (17.6) |
| **CLINICAL AND MEDICATION RELATED** | |
| **Monthly expenditure on drugs, Median (IQR)** | 50.0 (30.0, 100.0) |
| **Monthly expenditure on drugs** | |
| None/ Health insurance | 68 (20.7) |
| <50 cedis (<8.0 USD) | 101 (30.7) |
| 51–100 cedis (8.1–16.0 USD) | 88 (26.7) |
| >100 cedis (>16.0 USD) | 72 (21.9) |
| **Number of anti-hypertensive medications, Median (IQR)** | 2 (1, 3) |
| **Number of anti-hypertensive medications** | |
| <2 medicine | 98 (29.8) |
| 2+ medicines | 231 (70.2) |
| **Number of anti-diabetes medications, Median (IQR)** | 2.0 (1.0, 2.0) |
| **Number of anti-diabetes medications** | |
| <2 medicine | 161 (48.9) |
| 2+ medicines | 168 (51.1) |
| **Other medications** | |
| **Dyslipidaemia medications** | |
| No | 264 (80.2) |

*(Continued)*

**Table 1.** (Continued)

| Characteristics | Frequency (%) |
|---|---|
| Yes | 65 (19.8) |
| **Soluble aspirin** | |
| No | 314 (95.4) |
| Yes | 15 (4.6) |
| **Frequency of daily dose of medication** | |
| Once | 145 (44.1) |
| Twice | 181 (55.0) |
| Three times | 3 (0.9) |
| **Have co-morbidities** | |
| No | 269 (81.8) |
| Hyperlipidaemia | 57 (17.3) |
| Stroke | 3 (0.9) |
| **Family history of hypertension** | |
| Yes | 224 (68.1) |
| No | 105 (31.9) |
| **Family history of diabetes mellitus** | |
| Yes | 164 (49.8) |
| No | 165 (50.2) |
| **Duration since diagnosis of hypertension in years, Median (IQR)** | 5.0 (2.0, 6.0) |
| **Duration since diagnosis of hypertension** | |
| <5 years | 151 (45.9) |
| 5+ years | 178 (54.1) |
| **Duration since diagnosis of Diabetes Mellitus in years, Median (IQR)** | 3.0 (1.0, 5.0) |
| **Duration since diagnosis of Diabetes Mellitus** | |
| <5 years | 243 (73.9) |
| 5+ years | 86 (26.1) |
| **Frequency of follow-up** | |
| Every 2 weeks | 22 (6.7) |
| Monthly | 86 (26.1) |
| Every 2 months | 221 (67.2) |
| **Medication related burden** | |
| Minimum burden | 228 (69.3) |
| Moderate/high burden | 101 (30.7) |
| **Medication adherence** | |
| Non-adherence | 208 (63.2) |
| Adherence | 121 (36.8) |

SD: Standard deviation. IQR: Interquartile range. M: Multiple choice respond. USD: United state dollars conversion in November 2021.

Hyperlipidaemia and/or stroke co-morbidities were prevalent among 17.3% and 0.9% of the participants respectively. A medication-related burden was moderate/high among 30.7% whilst adherence to medication was 36.8% (Table 1).

## Blood pressure and blood glucose levels among participants

The median SBP was 140 mmHg (IQR: 128 to 157mmHg), DBP was 85 mmHg (IQR: 77 to 92mmHg) and blood glucose was 6.5 mmol/L (IQR: 5.6 to 8.3 mmol/L). (Fig 1).

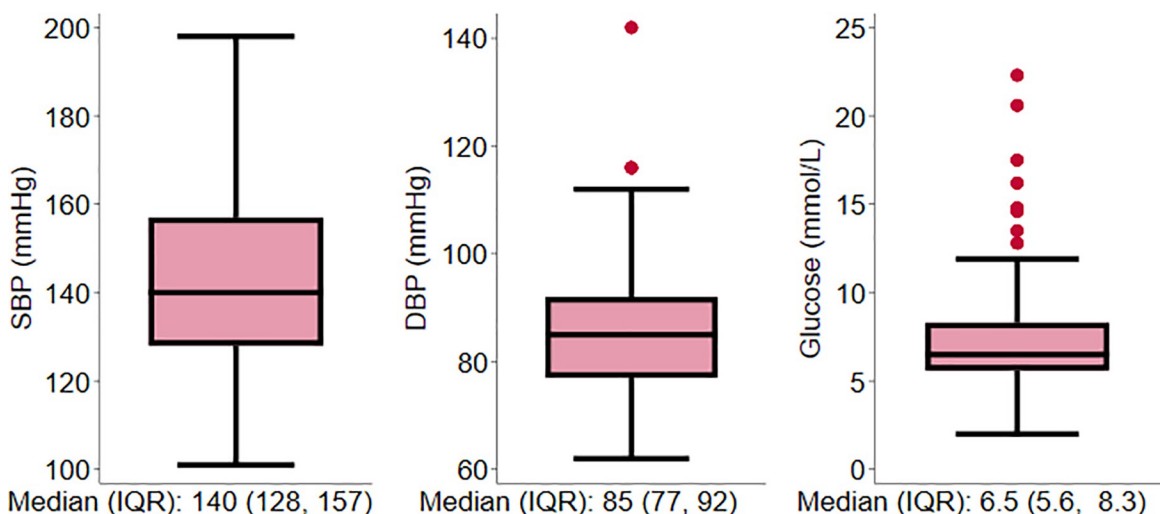

**Fig 1. Systolic blood pressure, diastolic blood pressure, and blood glucose levels among study participants.**

Less than half (41.3%) of the participants had controlled BP levels (SBP<140mmHg & DBP<90mmHg) (95% CI: 36.0% - 46.8%) whilst most (57.1%) had controlled blood glucose levels (<7.0 mmol/l) (95% CI: 51.7% - 62.4%). About a fifth (23.4%) had uncontrolled BP and blood glucose levels, 35.3% had uncontrolled blood pressure but controlled blood glucose levels, 19.5% had controlled BP levels but uncontrolled blood glucose levels and 21.8% had controlled BP and blood glucose levels (Table 2).

## Bivariate analysis of the association between characteristics of participants and outcomes

**Blood pressure control levels only.** The bivariate analysis showed that the statistically significant socio-demographic factors associated with the controlled levels of BP among the participants included age group (p = 0.024), marital status (p = 0.001) and the highest level of education (p = 0.001). The statistically significant clinical and medication-related factors associated with BP levels included expenditure on drugs (p = 0.014), number of anti-hypertensive medications (p<0.001), number of anti-diabetic medications (p<0.001), frequency of daily dose of medication (p<0.001), duration of hypertension diagnosis (p = 0.031), and frequency of follow-up visits to clinic (p = 0.021) (Table 3).

**Table 2. Blood pressure and blood glucose levels.**

| Outcomes | Frequency (N = 329) | Percentage | 95% CI |
|---|---|---|---|
| **Blood pressure level** | | | |
| Uncontrolled (SBP≥140mmHg or DBP ≥90mmHg) | 193 | 58.7 | [53.2, 63.9] |
| Controlled (SBP<140mmHg & DBP<90mmHg) | 136 | 41.3 | [36.1, 46.8] |
| **Blood glucose level** | | | |
| Uncontrolled blood glucose (7.0+ mmol/L) | 141 | 42.9 | [37.6, 48.3] |
| Controlled blood glucose (<7.0 mmol/L) | 188 | 57.1 | [51.7, 62.4] |
| **Blood pressure and blood glucose level** | | | |
| Uncontrolled BP & uncontrolled blood glucose | 77 | 23.4 | [19.1, 28.3] |
| Uncontrolled BP but controlled blood glucose | 116 | 35.3 | [30.3, 40.6] |
| Controlled BP but uncontrolled blood glucose | 64 | 19.5 | [15.5, 24.1] |
| Controlled BP & controlled blood glucose | 72 | 21.8 | [17.7, 26.7] |

**Table 3. Bivariate analysis of factors associated with blood pressure level and blood glucose level among hypertensive and type-2 diabetes mellitus co-morbid patients.**

| | Total | Controlled BP only | | Controlled blood glucose only | | Controlled BP and blood glucose level | |
|---|---|---|---|---|---|---|---|
| Variables & categories | N | n (%) | P-value | n (%) | P-value | n (%) | P-value |
| Overall | 329 | 136 (41.3) | | 188 (57.1) | | 72 (21.8) | |
| **SOCIO-DEMOGRAPHICS** | | | | | | | |
| **Sex** | | | 0.910 | | 0.001 | | 0.140 |
| Male | 144 | 60 (41.7) | | 68 (47.2) | | 26 (18.1) | |
| Female | 185 | 76 (41.1) | | 120 (64.9) | | 46 (24.9) | |
| **Age group** | | | 0.024 | | 0.023 | | 0.240 |
| <50 | 80 | 22 (27.5) | | 51 (63.7) | | 13 (16.3) | |
| 50–59 years | 108 | 46 (42.6) | | 67 (62.0) | | 29 (26.9) | |
| 60–69 | 78 | 36 (46.2) | | 33 (42.3) | | 14 (17.9) | |
| 70+ | 63 | 32 (50.8) | | 37 (58.7) | | 16 (25.4) | |
| **Marital status** | | | 0.001 | | 0.820 | | 0.140 |
| Single | 84 | 48 (57.1) | | 50 (59.5) | | 24 (28.6) | |
| Married | 192 | 65 (33.9) | | 107 (55.7) | | 35 (18.2) | |
| Divorced/widowed/separated | 53 | 23 (43.4) | | 31 (58.5) | | 13 (24.5) | |
| **Highest education** | | | 0.001 | | 0.006 | | <0.001 |
| No formal education | 59 | 33 (55.9) | | 36 (61.0) | | 20 (33.9) | |
| Primary | 72 | 23 (31.9) | | 40 (55.6) | | 13 (18.1) | |
| Secondary | 128 | 61 (47.7) | | 84 (65.6) | | 36 (28.1) | |
| Tertiary | 70 | 19 (27.1) | | 28 (40.0) | | 3 (4.3) | |
| **Employment status** | | | 0.340 | | 0.049 | | 0.004 |
| Unemployed/retired | 89 | 33 (37.1) | | 43 (48.3) | | 10 (11.2) | |
| Employed | 240 | 103 (42.9) | | 145 (60.4) | | 62 (25.8) | |
| **Monthly income** | | | 0.310 | | 0.011 | | 0.280 |
| 0–500 cedis (0–80 USD) | 150 | 61 (40.7) | | 93 (62.0) | | 38 (25.3) | |
| 501–1000 cedis (81.0–160 USD) | 121 | 46 (38.0) | | 72 (59.5) | | 25 (20.7) | |
| >1000 (>160 USD) | 58 | 29 (50.0) | | 23 (39.7) | | 9 (15.5) | |
| **CLINICAL AND MEDICATION RELATED** | | | | | | | |
| **Source of expenditure on drugs** | | | 0.014 | | 0.160 | | 0.092 |
| Health insurance only | 68 | 37 (54.4) | | 44 (64.7) | | 20 (29.4) | |
| Out-of-pocket | 261 | 99 (37.9) | | 144 (55.2) | | 52 (19.9) | |
| **Number of hypertensive medications** | | | <0.001 | | 0.015 | | <0.001 |
| <2 medicine | 98 | 63 (64.3) | | 46 (46.9) | | 34 (34.7) | |
| 2+ medicines | 231 | 73 (31.6) | | 142 (61.5) | | 38 (16.5) | |
| **Number of diabetes medications** | | | <0.001 | | 0.660 | | <0.001 |
| <2 medicine | 161 | 85 (52.8) | | 94 (58.4) | | 51 (31.7) | |
| 2+ medicines | 168 | 51 (30.4) | | 94 (56.0) | | 21 (12.5) | |
| **dyslipidaemia medications** | | | 0.970 | | 0.280 | | 0.023 |
| No | 264 | 109 (41.3) | | 147 (55.7) | | 51 (19.3) | |
| Yes | 65 | 27 (41.5) | | 41 (63.1) | | 21 (32.3) | |
| **Soluble Aspirin** | | | 0.240 | | 0.400 | | 0.036 |
| Does not take medication | 314 | 132 (42.0) | | 181 (57.6) | | 72 (22.9) | |
| Takes medication | 15 | 4 (26.7) | | 7 (46.7) | | 0 (0.0) | |
| **Frequency of daily dose of medication** | | | <0.001 | | 0.002 | | 0.460 |
| Once | 145 | 43 (29.7) | | 97 (66.9) | | 29 (20.0) | |
| Twice/thrice | 184 | 93 (50.5) | | 91 (49.5) | | 43 (23.4) | |

*(Continued)*

**Table 3.** (Continued)

| Variables & categories | Total | Controlled BP only | | Controlled blood glucose only | | Controlled BP and blood glucose level | |
|---|---|---|---|---|---|---|---|
| | N | n (%) | P-value | n (%) | P-value | n (%) | P-value |
| **Have co-morbidities** | | | 0.350 | | 0.026 | | 0.002 |
| None | 269 | 108 (40.1) | | 146 (54.3) | | 50 (18.6) | |
| Hyperlipidaemia /Stroke | 60 | 28 (46.7) | | 42 (70.0) | | 22 (36.7) | |
| **Family history of hypertension** | | | 0.530 | | 0.056 | | 0.570 |
| Yes | 224 | 90 (40.2) | | 136 (60.7) | | 51 (22.8) | |
| No | 105 | 46 (43.8) | | 52 (49.5) | | 21 (20.0) | |
| **Family history of diabetes mellitus** | | | 0.620 | | 0.950 | | 0.810 |
| Yes | 164 | 70 (42.7) | | 94 (57.3) | | 35 (21.3) | |
| No | 165 | 66 (40.0) | | 94 (57.0) | | 37 (22.4) | |
| **Duration since diagnosis of hypertension** | | | 0.031 | | 0.950 | | 0.430 |
| <5 years | 151 | 72 (47.7) | | 86 (57.0) | | 36 (23.8) | |
| 5+ years | 178 | 64 (36.0) | | 102 (57.3) | | 36 (20.2) | |
| **Duration since diagnosis of T2DM** | | | 0.095 | | 0.120 | | 0.960 |
| <5 years | 243 | 107 (44.0) | | 145 (59.7) | | 53 (21.8) | |
| 5+ years | 86 | 29 (33.7) | | 43 (50.0) | | 19 (22.1) | |
| **Frequency of follow-up** | | | 0.021 | | 0.007 | | 0.060 |
| 2 weeks/monthly | 108 | 35 (32.4) | | 73 (67.6) | | 17 (15.7) | |
| Every 2 month | 221 | 101 (45.7) | | 115 (52.0) | | 55 (24.9) | |
| **Medication related burden** | | | 0.200 | | 0.430 | | 0.046 |
| Minimum burden | 228 | 89 (39.0) | | 127 (55.7) | | 43 (18.9) | |
| Moderate/high burden | 101 | 47 (46.5) | | 61 (60.4) | | 29 (28.7) | |
| **Adherence to medication** | | | 0.810 | | 0.180 | | 0.210 |
| Non-adherence | 208 | 87 (41.8) | | 113 (54.3) | | 41 (19.7) | |
| Adherence | 121 | 49 (40.5) | | 75 (62.0) | | 31 (25.6) | |

**Blood glucose control levels only.** The bivariate analysis showed that sex (p = 0.001), age group (p = 0.023), highest education (p = 0.006), employment status (p = 0.049) and monthly income (p = 0.011) were the statistically significant socio-demographic factors associated with the blood glucose levels of participants. The statistically significant clinical and medication related factors associated with blood glucose level included number of anti-hypertensive medications (p = 0.015), frequency of daily dose of medication (p = 0.002), presence of hyperlipidaemia or stroke co-morbidity (p = 0.026) and frequency of follow-up visits to clinic (p = 0.007) (Table 3).

**Blood pressure and blood glucose control levels.** The bivariate analysis showed that highest level of education (p<0.001) and employment status (p = 0.004) were the only socio-demographic characteristics associated with controlled BP and blood glucose levels. Number of anti-hypertensive medications (p<0.001), number of anti-diabetic medication (p<0.001), dyslipidaemia medication (p = 0.023), soluble aspirin (p = 0.036), presence of hyperlipidaemia/stroke comorbidities (p = 0.002), and medication related burden (p = 0.046) were the clinical- and medication- related factors associated with controlled BP and blood glucose level (Table 3).

## Multivariable binary logistic regression model of factors associated with outcomes of the study

**Blood pressure control levels.** In the multivariable binary logistic regression model, the adjusted odds of controlled BP compared to those less than 50 years old was over 5 times high

among the 60–69 years old (AOR: 5.16, 95% CI: 1.77–15.02, p = 0.003) and over 9 times higher among the 70 years and older (AOR: 9.44, 95% CI: 3.06–29.16, p<0.001). Compared to the married, the odds of controlled BP were over 5 times high among the single (AOR: 5.70, 95% CI: 2.45–13.26, p = 0.003). Also, compared to those with no formal education, the odds of controlled BP was 86% less among those with primary school education (AOR: 0.14, 95% CI: 0.05–0.35, p<0.001) and 68% less among those with tertiary education (AOR: 0.32, 95% CI: 0.12–0.82, p = 0.018). The adjusted odds of controlled BP was over 5 times high among the employed (AOR: 5.47, 95% CI: 2.30–12.97, p<0.001) (Table 4).

The odds of controlled BP was 84% less among those on 2 or more anti-hypertensive medications (AOR: 0.16, 95% CI: 0.07–0.37, p<0.001) and 70% less among those on 2 or more anti-diabetic medications (AOR: 0.30, 95% CI: 0.15–0.59, p = 0.001). Participants who take two or three daily doses of medications had increased odds of having controlled BP compared to those taking a single dose (AOR: 7.00, 95% CI:2.88–17.01, p<0.001). The odds of controlled BP were 54% less among those who had been diagnosed with hypertension for at least 5 years (AOR: 0.46, 95% CI: 0.23–0.91, p = 0.027) (Table 4).

**Blood Glucose control levels.** In the multivariable binary logistic regression model, the odds of controlled blood glucose levels were 2 times higher among females compared to males (AOR: 2.04, 95% CI: 1.04–4.00, p = 0.038). The odds of controlled blood glucose levels were 50% less among the single compared to the married (AOR: 0.50, 95% CI: 0.27–0.96, p = 0.037). Also, participants with the highest monthly income levels (>1000 cedis) had 66% reduced odds of controlled blood glucose level (AOR: 0.34, 95% CI: 0.14–0.78, p = 0.012) (Table 4).

The odd of controlled blood glucose level was 2 times among those on 2 or more anti-hypertensive medications (AOR: 2.52, 95% CI: 1.34–4.74, p = 0.004). Participants who took two or three daily doses of medications had reduced odds of having controlled blood glucose levels compared to those taking a single dose (AOR: 0.52, 95% CI: 0.29–0.93, p<0.027). The odds of controlled blood glucose levels were 53% less among those with follow up visits every 2 months (AOR: 0.47, 95% CI: 0.26–0.85, p = 0.012) (Table 4).

**Both controlled blood pressure and blood glucose levels.** In the multivariable binary logistic regression model, the adjusted odds of controlled BP and blood glucose levels compared to those with no formal education was 84% less among primary school holders (AOR: 0.16, 95% CI: 0.05–0.49, p = 0.001) and 89% less among the tertiary school educated (AOR: 0.11, 95% CI: 0.03–0.44, p = 0.002). The adjusted odds of controlled BP and blood glucose were 10 times high among the employed (AOR: 10.12, 95% CI: 3.28–31.25, p<0.001). Increased monthly income level was associated with decreased odds of controlled BP and blood glucose level, thus compared to those with 0–500 cedis of monthly income, the odds of controlled BP and blood glucose were 75% less for those with 501–1000 cedis monthly income (AOR: 0.25, 95% CI: 0.10–0.60, p = 0.002) and 93% less among those earning more than 1000.00 cedis (AOR: 0.07, 95% CI: 0.02–0.25, p<0.001) (Table 4).

The odds of controlled BP and blood glucose level was 56% less among those whose expenditure on medications were out-of-pocket (AOR: 0.44, 95% CI: 0.20–0.98, p = 0.046). Compared to those on single medications, the odds of controlled BP and blood glucose was 60% less among those taking 2 or more antihypertensive medications (AOR: 0.40, 95% CI: 0.17–0.94, p = 0.036) and 73% less among those taking 2 or more anti-diabetic medications (AOR: 0.27, 95% CI: 0.13–0.56, p<0.001). The odds of controlled BP and blood glucose were over 3 times high among those having co-morbid hyperlipidaemia/ stroke (AOR: 3.35, 95% CI: 1.50–7.48, p = 0.003). Controlled BP and blood glucose levels were significantly high among participants with moderate or high medication related burden (AOR: 2.42, 95% CI: 1.17–5.05, p = 0.018). Adherence to medication was associated with over 2 times higher odds of having both controlled BP and blood glucose level (AOR: 2.12, 95% CI:1.03, 4.34, p = 0.040) (Table 4).

**Table 4. Multivariable binary logistic regression model of factors associated with controlled levels of blood pressure, glucose and both blood pressure and glucose among hypertensive and type 2 diabetes mellitus co-morbid patients.**

| | Multiple binary logistic regression model | | | | | |
| --- | --- | --- | --- | --- | --- | --- |
| | Controlled blood pressure level | | High blood glucose level | | Controlled Blood pressure and blood glucose level | |
| Variables and categories | AOR [95% CI] | P-value | AOR [95% CI] | P-value | AOR [95% CI] | P-value |
| **SOCIO-DEMOGRAPHICS** | | | | | | |
| **Sex** | | | | | | |
| Male | 1.00 [reference] | | 1.00 [reference] | | 1.00 [reference] | |
| Female | 0.93 [0.47, 1.84] | 0.828 | 2.04 [1.04, 4.00] | 0.038 | 1.31 [0.49, 3.48] | 0.588 |
| **Age group** | | | | | | |
| <50 | 1.00 [reference] | | 1.00 [reference] | | 1.00 [reference] | |
| 50–59 years | 0.97 [0.37, 2.56] | 0.952 | 1.38 [0.65, 2.92] | 0.401 | 1.09 [0.36, 3.35] | 0.877 |
| 60–69 | 5.16 [1.77, 15.02] | 0.003 | 0.65 [0.27, 1.55] | 0.331 | 1.25 [0.31, 5.00] | 0.753 |
| 70+ | 9.44 [3.06, 29.16] | <0.001 | 1.22 [0.37, 3.99] | 0.743 | 2.33 [0.66, 8.20] | 0.188 |
| **Marital status** | | | | | | |
| Married | 1.00 [reference] | | 1.00 [reference] | | 1.00 [reference] | |
| Single | 5.70 [2.45, 13.26] | <0.001 | 0.50 [0.27, 0.96] | 0.037 | 0.88 [0.41, 1.88] | 0.736 |
| Divorced/widowed/separated | 2.25 [0.96, 5.29] | 0.063 | 0.91 [0.35, 2.33] | 0.837 | 1.97 [0.78, 4.98] | 0.151 |
| **Highest education** | | | | | | |
| No formal education | 1.00 [reference] | | 1.00 [reference] | | 1.00 [reference] | |
| Primary | 0.14 [0.05, 0.35] | <0.001 | 0.48 [0.21, 1.09] | 0.081 | 0.16 [0.05, 0.49] | 0.001 |
| Secondary | 0.68 [0.30, 1.56] | 0.361 | 1.81 [0.88, 3.71] | 0.104 | 0.82 [0.28, 2.34] | 0.705 |
| Tertiary | 0.32 [0.12, 0.82] | 0.018 | 0.58 [0.24, 1.40] | 0.226 | 0.11 [0.03, 0.44] | 0.002 |
| **Employment status** | | | | | | |
| Unemployed/retired | 1.00 [reference] | | 1.00 [reference] | | 1.00 [reference] | |
| Employed | 5.47 [2.30, 12.97] | <0.001 | 1.88 [0.79, 4.50] | 0.156 | 10.12 [3.28, 31.25] | <0.001 |
| **Monthly income** | | | | | | |
| 0–500 cedis (0–80 USD) | 1.00 [reference] | | 1.00 [reference] | | 1.00 [reference] | |
| 501–1000 cedis (81.0–160 USD) | 0.64 [0.32, 1.28] | 0.208 | 0.83 [0.43, 1.59] | 0.573 | 0.25 [0.10, 0.60] | 0.002 |
| >1000 (>160 USD) | 0.75 [0.28, 2.01] | 0.564 | 0.34 [0.14, 0.78] | 0.012 | 0.07 [0.02, 0.25] | <0.001 |
| **CLINICAL AND MEDICATION RELATED** | | | | | | |
| **Source of expenditure on drugs** | | | | | | |
| Health insurance only | 1.00 [reference] | | 1.00 [reference] | | 1.00 [reference] | |
| Out-of-pocket | 0.68 [0.33, 1.41] | 0.302 | 0.56 [0.28, 1.11] | 0.095 | 0.44 [0.20, 0.98] | 0.046 |
| **Number of hypertensive medications** | | | | | | |
| <2 medicine | 1.00 [reference] | | 1.00 [reference] | | 1.00 [reference] | |
| 2+ medicines | 0.16 [0.07, 0.37] | <0.001 | 2.52 [1.34, 4.74] | 0.004 | 0.40 [0.17, 0.94] | 0.036 |
| **Number of diabetes medications** | | | | | | |
| <2 medicine | 1.00 [reference] | | 1.00 [reference] | | 1.00 [reference] | |
| 2+ medicines | 0.30 [0.15, 0.59] | 0.001 | 0.79 [0.45, 1.39] | 0.419 | 0.27 [0.13, 0.56] | <0.001 |
| **Frequency of daily dose of medication** | | | | | | |
| Once | 1.00 [reference] | | 1.00 [reference] | | 1.00 [reference] | |
| Twice/thrice | 7.00 [2.88, 17.01] | <0.001 | 0.52 [0.29, 0.93] | 0.027 | 2.40 [0.92, 6.25] | 0.073 |
| **Have co-morbidities** | | | | | | |
| None | 1.00 [reference] | | 1.00 [reference] | | 1.00 [reference] | |
| Hyperlipidaemia /Stroke | 1.06 [0.38, 2.92] | 0.911 | 2.40 [0.96, 5.99] | 0.061 | 3.35 [1.50, 7.48] | 0.003 |
| **Duration since diagnosis of hypertension** | | | | | | |
| <5 years | 1.00 [reference] | | 1.00 [reference] | | 1.00 [reference] | |
| 5+ years | 0.46 [0.23, 0.91] | 0.027 | 1.70 [0.92, 3.12] | 0.089 | 1.29 [0.64, 2.58] | 0.474 |

*(Continued)*

**Table 4.** (Continued)

| | Multiple binary logistic regression model | | | | | |
|---|---|---|---|---|---|---|
| | Controlled blood pressure level | | High blood glucose level | | Controlled Blood pressure and blood glucose level | |
| Variables and categories | AOR [95% CI] | P-value | AOR [95% CI] | P-value | AOR [95% CI] | P-value |
| **Duration since diagnosis of Diabetes Mellitus** | | | | | | |
| <5 years | 1.00 [reference] | | 1.00 [reference] | | 1.00 [reference] | |
| 5+ years | 0.42 [0.16, 1.07] | 0.069 | 0.75 [0.37, 1.52] | 0.424 | 1.75 [0.78, 3.95] | 0.177 |
| **Frequency of follow-up** | | | | | | |
| 2 weeks/monthly | 1.00 [reference] | | 1.00 [reference] | | 1.00 [reference] | |
| Every 2 month | 1.94 [0.99, 3.79] | 0.054 | 0.47 [0.26, 0.85] | 0.012 | 1.86 [0.83, 4.19] | 0.132 |
| **Medication related burden** | | | | | | |
| Minimum burden | 1.00 [reference] | | 1.00 [reference] | | 1.00 [reference] | |
| Moderate/high burden | 1.37 [0.73, 2.57] | 0.328 | 2.77 [1.59, 4.85] | <0.001 | 2.42 [1.17, 5.05] | 0.018 |
| **Adherence to medications** | | | | | | |
| Non-adherence | 1.00 [reference] | | 1.00 [reference] | | 1.00 [reference] | |
| Adherence | 0.99 [0.56, 1.75] | 0.983 | 1.65 [0.96, 2.87] | 0.071 | 2.12 [1.03, 4.34] | 0.040 |

AOR: Adjusted odds ratio. CI: Confidence interval.

## Discussion

The current study sought to assess the BP and blood glucose levels among persons with co-morbid hypertension and T2DM, and the factors associated with these outcomes. The research observed that less than half (41.3%) of patients with hypertension and diabetes comorbidity had their BP under control. These findings are similar to findings a from South African study (42%) [26] and the Jimma University Medical Center (43.51%) [4]. Though BP control levels from this research is better compared to other studies conducted in Addis Ababa (19.4%) [27], and Malaysia (23.5%) [28], this was relatively lower than a study conducted in the Ho municipality in Ghana which reported BP control of 58.7% among people living with diabetes. Furthermore, this study showed the median SBP and DBP were 140 mmHg (IQR: 128–157) and 85 mmHg (IQR: 77–92) respectively. The SBP and DBP estimates from this study were higher than the 135.4 mmHg and 83.3 mmHg estimates from a study among out-patients in two diabetic clinics in Ghana [29]. However, recommendations for managing BP, such as the JNC 8, typically call for lower systolic and diastolic levels in diabetics [12]. Maintaining sufficient BP control is the primary therapeutic goal; hence these findings point to the need for greater attention to the optimum care of patients with co-morbid hypertension and T2DM. Also, in line with the objectives of the Ghana NCD policy, which aims to strengthen early detection and management of NCDs including co-morbid hypertension and T2DM, maintaining BP and blood glucose control will lead to a reduction in morbidity and mortality from NCDs [19,24].

Compared with some patients with diabetes in Ghana, the blood glucose levels in this study was higher [30–32]. Generally, in low-resourced settings like Ghana, and especially in public health facilities where this study was conducted, glycaemic control was assessed using the fasting blood sugar. This is due to the high costs associated with more robust measures like the glycated haemoglobin (HbA1c) which measures average glycaemia over three months.

Also, a fifth of the participants had uncontrolled BP and blood glucose levels whilst another one-fifth had controlled levels for both BP and blood glucose. Without effort from patients, achieving target BP and fasting blood glucose (FBG) will be a difficult challenge. However,

when patients ensure optimal adherence to recommended treatment, which can also be based on patients' adherence to dietary and lifestyle changes. In cases when the initial aims are not met, it is advised to alter the medicines and to regularly evaluate patients [33]. Additionally, it has been demonstrated that following clinical guidelines when prescribing drugs for T2DM and hypertension improves clinical outcomes [34].

The findings from this study also showed that controlled BP levels were high among the older age groups which is inconsistent with the findings from a similar study conducted in Ethiopia where patients who fell among the older age groups had two times uncontrolled BP compared to patients among the younger age groups [4] This observed result may be due to the fear in the prevalence of worse clinical outcomes in the elderly who present with comorbid hypertension and diabetes and hence a better rate of adherence to treatment. Although other studies did not find any significant association between marital status and control of BP levels [4,35], this study showed that those who were not currently married had controlled BP levels. Patients taking two or three doses of medications daily, had controlled BP levels which is inconsistent with other studies which have showed that multiple daily dosing of medications leads to a higher rate of non-adherence and consequently unfavourable clinical outcomes [36–38]. The differences in results may be due to a better understanding of the co-morbid nature of their disease condition and therefore a better rate of adherence to their treatment. This study further revealed that those who were employed had controlled BP levels which disagrees with a similar study in Addis Ababa where patients who were employed had uncontrolled BP levels [35]. This may be due to the strains of their jobs and therefore a recognition of the need to adhere to treatment to prevent a worsened clinical outcome. On the other hand, the study findings showed that controlled BP levels were low among those with some form of formal education. This disagrees with studies that have shown formal education to have a positive association with BP levels [4,35]. The differences in observed results may be due to the differences in health seeking behaviours. Participants taking 2 or more different anti-hypertensive and anti-diabetic medications had controlled BP levels which is inconsistent with other studies which have shown that patients who take two or more different medications for their chronic diseases feel burdened and therefore do not adhere to their treatment [36–38]. The observed results may be due to an understanding of the need to adhere to treatment. This study also revealed that those who have been diagnosed with hypertension for at least 5 years had controlled BP levels which is inconsistent with studies which have shown that patients who have been diagnosed with hypertension for over 5 years have uncontrolled BP levels due to a reduction in their health-seeking behaviour [35].

In terms of blood glucose control, the study findings agree with other studies which have shown that two or more medications, and moderate to high medication related burden were significantly associated poor blood glucose control levels [39,40]. Patients taking two or more medications may feel burdened by the number of pills they have to take which consequently leads to non-adherence to treatment and poor blood glucose control outcome [36]. On the other hand, although other studies disagree with the study findings which showed that two or three daily doses of medications were associated better blood glucose control levels, they agree that bi-monthly clinic follow-up periods were implicated blood glucose control levels [36,39,40]. The observed results may be due to an understanding of the disease nature and therefore an increase in health seeking behaviour. Though this study showed that being married and being female has a significant association with increased odds of blood glucose control levels and increased monthly income has significant association with decreased odds of blood glucose control levels, other studies found no association with blood control levels [39,40].

In terms of combined BP and blood glucose control, this study further found that being employed, having hyperlipidaemia or stroke co-morbidity, and moderate to high medication related burden were associated with high controlled BP and blood glucose levels. This is inconsistent with other studies which have shown that being employed, having other co-morbidities and moderate to high medication related burden are associated with uncontrolled BP and blood glucose levels [35,39,40]. In order to prevent unfavourable outcomes due to the presence of other co-morbidities and the strains of their jobs, these patients may have adhered to their treatment regimen which is probably the reason for the observed results. On the other hand, this study showed that having some form of formal education is associated with decreased chances of having both BP and blood glucose levels controlled, which is inconsistent with studies which have shown significant association between formal education and controlled levels of BP and blood glucose [4,35]. This may be due to a decrease in health seeking behaviour. Consistent with other studies [36–38], this study has also shown two or more anti-hypertensive and anti-diabetic medications to be associated with decreased chances of having both BP and blood glucose levels controlled. This is due to the pill burden these patients feel which may lead to non-adherence to treatment and a deterioration of their condition. Socio-behavioural interventions may be needed for encouraging such patients to adhere to their medications. Although other studies found no association between income and BP and blood glucose, this study showed that increased monthly income level and out-of-pocket payment for medications were associated with decreased chances of having both BP and blood glucose control levels and these have implications for medicines availability, affordability and adherence behaviour [4,35].

## Study limitations

This study is a cross-sectional study hence cannot establish causal inferences but rather associations. Interpretation of findings should therefore be done cautiously. Since this was a hospital-based study, information on the estimates of BP levels and blood glucose were facility specific and may not reflect what pertains within the general community exhibit. Also, because this study measured the BP and blood glucose level of participants at one time point during the interview, it is unable to determine the fluctuations at different time points to establish consistencies in these parameters. Nonetheless, this study provides some estimates on the extent of BP and blood glucose controls among patients with hypertension and T2DM to inform the development of interventions. Again, the rigorous nature of the analytical procedure also reduces the potential biases that are likely to occur. The sample size of 339 is also large enough to draw valid conclusions from a single centre study.

## Conclusion

In this study, two in five patients with hypertension and T2DM had controlled BP levels, three in five had controlled blood glucose levels and one in five had both BP and blood glucose levels under control with factors such as employment status, hyperlipidaemia or stroke co-morbidity, adherence to medication, and moderate to high medication related burden being associated with the control. Identifying patients at risk of poor BP and blood glucose control can lead to targeted interventions in line with the Ghana NCD policy on strengthening management especially among high-risk groups to reduce morbidity and mortality from NCDs. In addition, policy makers should institute measures to ensure that health promotion and education on long-term diseases such as hypertension and T2DM form part of the routine management and care practices. Potential drawbacks to poor control of BP and blood glucose

parameters should also be included in these policy documents to guide clinical practice and ensure better health outcomes for patients with co-morbid hypertension and T2DM.

## Acknowledgments

The authors would like to thank the study facilities and participants for their role in this study as well as Prof. Robert Horne and Prof. Janet Krska for the use of the MARS-5 and the LMQ tools respectively.

## Author Contributions

**Conceptualization:** Yakubu Alhassan, Emmanuella Baah-Nyarkoh, Irene A. Kretchy.

**Data curation:** Yakubu Alhassan, Emmanuella Baah-Nyarkoh, Irene A. Kretchy.

**Formal analysis:** Yakubu Alhassan.

**Investigation:** Emmanuella Baah-Nyarkoh.

**Methodology:** Yakubu Alhassan, Emmanuella Baah-Nyarkoh, Irene A. Kretchy.

**Writing – original draft:** Yakubu Alhassan, Adwoa Oforiwaa Kwakye, Andrews K. Dwomoh, Irene A. Kretchy.

**Writing – review & editing:** Adwoa Oforiwaa Kwakye, Andrews K. Dwomoh, Vincent Jessey Ganu, Bernard Appiah, Irene A. Kretchy.

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
