## [Decision Letter · Decision Letter 0]

28 Sep 2022

PGPH-D-22-01382

Determinants of blood pressure and blood glucose control in patients with co-morbid hypertension and type 2 diabetes mellitus in Ghana: A hospital-based cross-sectional study.

Dear Dr. Kwakye,

Thank you for submitting your manuscript to PLOS Global Public Health. After careful consideration, we feel that it has merit but does not fully meet PLOS Global Public Health’s publication criteria as it currently stands. Therefore, we invite you to submit a revised version of the manuscript that addresses the points raised during the review process.

We look forward to receiving your revised manuscript.

Kind regards,

Tilahun Haregu

Academic Editor

Journal Requirements:

2. Please provide your detailed Financial Disclosure statement. This is published with the article. It must therefore be completed in full sentences and contain the exact wording you wish to be published.

a. Please clarify all sources of funding (financial or material support) for your study. List the grants (with grant number) or organizations (with url) that supported your study, including funding received from your institution. 

b. State the initials, alongside each funding source, of each author to receive each grant.

c. State what role the funders took in the study. If the funders had no role in your study, please state: “The funders had no role in study design, data collection and analysis, decision to publish, or preparation of the manuscript.”

d. If any authors received a salary from any of your funders, please state which authors and which funders.

3. Please provide separate figure files in .tif or .eps format only and remove any figures embedded in your manuscript file. Please also ensure that all files are under our size limit of 10MB.

Additional Editor Comments (if provided):

Reviewers' comments:

Reviewer's Responses to Questions

**Comments to the Author**

1. Does this manuscript meet PLOS Global Public Health’s publication criteria? Is the manuscript technically sound, and do the data support the conclusions? The manuscript must describe methodologically and ethically rigorous research with conclusions that are appropriately drawn based on the data presented.

Reviewer #1: Yes

Reviewer #2: Yes

2. Has the statistical analysis been performed appropriately and rigorously?

Reviewer #1: Yes

Reviewer #2: Yes

3. Have the authors made all data underlying the findings in their manuscript fully available (please refer to the Data Availability Statement at the start of the manuscript PDF file)?

Reviewer #1: Yes

Reviewer #2: Yes

4. Is the manuscript presented in an intelligible fashion and written in standard English?

Reviewer #1: Yes

Reviewer #2: Yes

5. Review Comments to the Author

Reviewer #1: The manuscript is scientifically sound and the author has described the methods well albeit with a few revisions needed. The statistical analysis is well done and presented well however there is a slight labelling error in table 4.The conclusions are drawn for the data and i particularly like the author is aware of the limitations of his study. It should be noted that a good number authors results do not match what has been found in other studies and the author has given possible explanations and also recommended the need for a multi-center study. The author has good commend of english and sentecnes are well constructed and understood

Reviewer #2: 1) In "Study Design and Context" subsection, please clarify if the facility is a public or a private health facility. Also provide some information regarding whether Ghana has an NCD control program in place or not. Please mention the services that are available - eg. identification, management, follow-up services, monitoring of clinical status, etc - and whether services such as testing and provision of medicines are available to patients free of cost or not.

2) In the Results section, Table 4 has some contradictory findings between the "controlled BP group" and "controlled blood glucose group" (eg. patients who take 2 or 3 daily doses of medication had better odds of BP control compared to single daily dose, while the opposite was true for blood glucose control). Can these differences be explained?

3) In the Discussion, please provide some information about the status of the NCD program in Ghana and in the health facility in particular. In particular, are there any implementation challenges in the program which may affect the treatment services and ultimately affect the control of BP and blood glucose among patients? For example, if availability of medications is frequently disrupted, patients may have to go to private health facilities or pharmacies to buy their medication, which would lead to higher out of pocket expenditure and affect their control of BP and blood glucose. Are all the commonly prescribed antihypertensives and oral hypoglycaemic agents included in the national essential drugs list? If not, that would again lead to high out of pocket expenditure. Does the health facility run out of blood sugar testing kits? If yes, that would adversely affect testing services.

4) In the Limitations section, do mention that since the study focused on the patients who were already diagnosed and seeking treatment at the facility, this may not reflect the actual proportion of controlled BP and blood glucose in the community, since it is quite likely that there is a considerable number of patients with hypertension and diabetes, who are not yet diagnosed.

5) Also, since the study is focused on the control of BP and blood glucose, it would have been ideal if the adherence to treatment was also probed.

6) In the Conclusion section, I would suggest adding some more recommendations for policymakers - the factors which are associated with decreased control of BP/blood sugar could be useful in identifying vulnerable groups, and their adherence to medication and hence level of control could potentially be improved by targeted IEC campaigns in the community by community health workers. Also, better coordination between levels of care (such as between facility level health workers and community health workers) could result in better patient outcomes (by ensuring improved adherence to treatment, improved health seeking behaviour, etc).

6. PLOS authors have the option to publish the peer review history of their article (what does this mean?). If published, this will include your full peer review and any attached files.

**Do you want your identity to be public for this peer review?** For information about this choice, including consent withdrawal, please see our Privacy Policy.

Reviewer #1: No

Reviewer #2: No

---

## [Editor Report · Decision Letter 1]

11 Nov 2022

Determinants of blood pressure and blood glucose control in patients with co-morbid hypertension and type 2 diabetes mellitus in Ghana: A hospital-based cross-sectional study.

PGPH-D-22-01382R1

Dear Dr. Kwakye,

We are pleased to inform you that your manuscript 'Determinants of blood pressure and blood glucose control in patients with co-morbid hypertension and type 2 diabetes mellitus in Ghana: A hospital-based cross-sectional study.' has been provisionally accepted for publication in PLOS Global Public Health.

Best regards,

Tilahun Haregu

Academic Editor
